# Resonance-Based Microwave Technique for Body Implant Sensing

**DOI:** 10.3390/s19224828

**Published:** 2019-11-06

**Authors:** Giselle González-López, Lluis Jofre Roca, Susana Amorós García de Valdecasas, Oriol Rodríguez-Leor, Carolina Gálvez-Montón, Antoni Bayés-Genís, Joan O’Callaghan

**Affiliations:** 1School of Telecommunication Engineering, Universitat Politècnica de Catalunya, 08034 Barcelona, Spain; luis.jofre@upc.edu (L.J.R.); susana.amoros@upc.edu (S.A.G.d.V.);; 2CIBERCV, Instituto de Salud Carlos III, 28029 Madrid, Spain; oriolrodriguez@gmail.com (O.R.-L.); cgalvez@igtp.cat (C.G.-M.); abayesgenis@gmail.com (A.B.-G.); 3Heart Institute (iCor), Germans Trias i Pujol University Hospital, 08916 Badalona, Spain; 4Department of Medicine, Universitat Autònoma de Barcelona, 08193 Barcelona, Spain; 5ICREC Research Program (Health Science Research Institute Germans Trias i Pujol), Can Ruti Campus, 08916 Badalona, Spain

**Keywords:** microwave sensing, biosensing, object localization, implant, differential resonance, stent, non-ionizing, phantom, relative permittivity

## Abstract

There is an increasing need for safe and simple techniques for sensing devices and prostheses implanted inside the human body. Microwave wireless inspection may be an appropriate technique for it. The implanted device may have specific characteristics that allow to distinguish it from its environment. A new sensing technique based on the principle of differential resonance is proposed and its basic parameters are discussed. This technique allows to use the implant as a signal scattering device and to detect changes produced in the implant based on the corresponding change in its scattering signature. The technique is first tested with a canonic human phantom and then applied to a real in vivo clinical experiment to detect coronary stents implanted in swine animals.

## 1. Introduction

There is an increasing use of body tiny implants and human prostheses known as implanted devices (ID) with varying geometries and compositions [1], but having in common dimensions up to some tens of millimeters that when used in a biological environment with permittivities around the 50 s may have an electromagnetic resonance into the UHF (0.3–3.0 GHz) frequency band.

These IDs may change some characteristics over time in such a way that their functionality may be compromised. These changes may be triggered by wear corrosion, breakdowns, adherence of biological tissues, etc. This would most likely result in operational malfunctioning and, in most cases, a change in its electrical parameters, which will result in a change of its resonant frequency. Examples of these IDs may include metal stents (usually covered by drug-eluding substances) [2], which currently are the preferred solution for the treatment of IHD (Ischemic Heart Disease) [3], pins and screws in orthopedics and craniofacial surgery, and artificial joints [4,5], with typical lengths ranging from 1 to 30 cm. These, when implanted in regions of the human body with permittivities between 4 and 57, produce a resonance in the UHF region.

UHF signals with a convenient dynamic range and appropriate antenna probes [6] are able to penetrate some tens of cm into the body and interact with those devices giving a scattering response that may be strongly influenced by the electrical resonance of the ID, therefore, allowing to monitor the state of the device. In order to obtain a proper diagnostic, it is necessary to have a clear signature of the scattered signal, as the one produced by a change of the resonant frequency.

Current solutions for pre-clinical identification of some of the previously mentioned conditions are either invasive, ionizing or require complex procedures. This is the case of intravascular ultrasound (IVUS) [7], X-ray angiography (XRA) [8] or intravascular optical coherence tomography (OCT) [9]. High-frequency solutions have been within the scope of many studies, however, in most cases, they are just able to detect superficial implants [10] or require the previous incorporation in the implant of an electronic chip [11].

In this paper, we propose to use implants as signal scattering devices and to detect any physical variation produced in the implant based on the corresponding variation of its scattering signature. The presented technique is able to extract the signature contained in the resonance frequency of the backscattered signal produced by the target device. We show that this frequency depends on the electrical length of the device and may be used to detect, position and monitor changes in devices having electrical dimensions in the range of cm and depths of several centimetres.

In Section 2, we present the analytic formulation of the presented scenario and the technique to extract the resonance. In Section 3, we apply the technique to a body phantom hexagonal geometry, by means of a monostatic or a bistatic scattering measurement. A parametric analysis in terms of the IFFT (Inverse Fast Fourier Transform) of the detection capabilities of the proposed technique is then performed. In Section 4, we apply the technique to a coronary stent implanted in a swine animal. Finally, in Section 5, some conclusive remarks are presented.

## 2. Resonant Scattering Produced by the Implanted Device (ID)

In this section, the basic diagram and expressions supporting the resonance-based monitoring principle are presented. The ID is modeled, without a loss of generality, as a generic electric dipole-based antenna with a virtual port defined somewhere in its structure from which a resonance frequency may be defined.

The generic scenario, shown in Figure 1, consists of a bistatic geometry with the transmitting (Tx) and receiving (Rx) probe antennas represented by two dielectrically filled ridge horn antennas [6], and the ID is represented as a cylindrical object of length $l_{ID}$, located at a distance ${\vec{r}}_{TD}$ and ${\vec{r}}_{RD}$ from Tx and Rx probes respectively, all of it immersed into a dielectric medium of permittivity $\varepsilon_{r}\left(r\right)$, supposed to be low frequency dispersive in the range of the ID resonance. Let $P_{T}={\left|a_{T}\right|}^{2}$ and $P_{R}={\left|b_{R}\right|}^{2}$ be the transmitted and received power into the Tx and Rx respectively, $Z_{in}^{ID}$ the impedance of the ID when fed into a virtual port, and $Z_{LD}^{ID}$ a virtual impedance loading the device’s port, that will model the operational state of the ID.

In a complex scenario, as those found in real situations, far-field conditions and homogeneous-space propagation do not meet, and the conventional radar formulation is not applicable. In consequence, an approach based on the reciprocity theorem is needed [12]. The signal $b_{R}$ measured into the receiving antenna may be expressed as [13]:(1)$$\begin{array}{cc} \rho_{TR}^{ID} & =\frac{b_{R}}{a_{R}}=\frac{V_{OC}^{TD}V_{OC}^{RD}}{2P_{T}}\frac{1}{Z_{in}^{ID}+Z_{LD}^{ID}}\\ \; & =\frac{h_{eff}^{2}E_{inc}^{TD}E_{inc}^{RD}}{2P_{T}}\frac{1}{Z_{in}^{ID}+Z_{LD}^{ID}} \end{array}$$
where $V_{OC}^{TD}$ and $V_{OC}^{RD}$ are the open-ended circuit voltages at the port of the ID when illuminated by the Tx or Rx (reciprocity-based) antennas that may respectively be expressed as the product of the incident field $E_{inc}^{TD}$ and $E_{inc}^{RD}$ into the ID produced by the Tx and Rx (reciprocity-based) and the ID effective high ($h_{eff}$).

For the prove of concept of the principle we will consider a cylindrical electric dipole (thin conducting cylinder) of length $l_{ID}$, with its first resonant frequency (half wave dipole) at $f_{res}^{ID}$. The input impedance $Z_{in}^{ID}$ may be approached, provided than the Tx/Rx antennas are significantly distant and/or the medium has a certain amount of loss as it would be the case for most of the media of interest, by its self-impedance $Z_{11}^{ID}$. It is then possible to express the self-impedance of the electric dipole immersed into a medium of permittivity $\varepsilon_{r}$ at a frequency *f* around its resonance frequency ($f_{res}^{ID}$), as [14]:(2)$$\begin{array}{cc} Z_{in}^{ID} & ≅Z_{11}^{ID}=\left(73+j43\frac{\frac{fl_{ID}\sqrt{\varepsilon_{r}}}{c_{0}}-0.45}{0.05}\right)\Omega \end{array}$$
where $c_{0}$ is the velocity of light in vacuum, and for which $Z_{in}^{ID}=73+j0$ at the resonance frequency, where a minimum into the input impedance is obtained. Substituting (Equation 2) into (Equation 1) we may then calculate the frequency response of the system, $\rho_{TR}^{ID}\left(f\right)$ for which a resonant maximum is obtained at the resonance, $f_{res}^{ID}$. The representation of this frequency response will give us the necessary information to diagnose the actual state of the resonance and, in consequence, the state of the implant.

As a first analytic approach, a cylindrical implant with a length $l_{ID}=$ 20 mm and distances $r_{TD}=r_{RD}=r_{TR}=r_{TRD}=$ 50 mm into a uniform medium of permittivity $\varepsilon_{r}=$ 50 with a frequency span from 0.3 to 1.2 GHz is considered. In this example, the Tx and Rx antennas are supposed to be broad-band biconical dipoles (with a radiation resistance $R_{a}^{TR}=75\;\Omega$ and a broad flat frequency covering the range of the ID resonance $f_{res}^{ID}$, while the implant itself resonates at a frequency of 0.85 GHz. In this specific example, because the far-field equations may be used (uniform medium and far-field distances), the expression for $\rho_{TR}^{ID}\left(f\right)$ may be analytically written as:(3)$$\begin{array}{cc} \rho_{TR}^{ID} & ≅-\frac{25l_{ID}^{2}}{R_{a}^{TR}\varepsilon_{r}}{\left(\frac{e^{-j21f\left(GHz\right)\sqrt{\varepsilon_{r}}r_{TRD}}}{r_{TRD}}\right)}^{2}\\ \times & \frac{1}{1+j\left(40f\left(GHz\right)\sqrt{\varepsilon_{r}}l_{ID}-5.3\right)} \end{array}$$

In a similar way $\rho_{TR}^{DR}$, the signal traveling directly from the Tx to the Rx that may be calculated with the Friis equation [15] into the homogeneous background medium.

Figure 2 represents the frequency response, between 0.3 and 1.2 GHz where a single resonance may be considered, of the Tx-ID-Rx system through the relation, $\rho_{TR}^{ID}\left(f\right)$, for two states: (a) short-circuited ($Z_{LD}^{ID}=$ 0.0), where the ID is supposed to work properly as a 20 mm cylindrical continuous element, and (b) open-circuited ($Z_{LD}^{ID}=$ 1000), where the ID is supposed to be broken in the middle as two shorter 10 mm elements in which the actual value of 10 mm represents the remaining structural backscattering.

For a real application, as it is seen in the next section, the measured magnitude will be the superposition of two signals: (a) the signal Tx-ID-Rx, with (b) the direct signal Tx-Rx, resulting into a differential pattern ($\rho_{TR}^{ID}-\rho_{TR}^{DR}$) in which resonance will appear as a minimum (Figure 2).

## 3. Numerical and Experimental Validation

The technique described in the previous section for difference resonance sensing of IDs in a biological material is tested first through simulations conducted in CST Studio, and then through experimental measurements conducted with two ridged horn antennas [6], integrated in the measurement set up in Figure 3 and filled with a phantom mimicking the permittivity of biological tissue which values have been extracted from [16].

This hexagonal set up has been manufactured in PLA (polylactic acid) using 3D printing technology to perfectly host the horns. The hexagonal shape allows to recreate the layout the antennas would have in an in vivo scenario. The dimensions of the hexagonal box are: inner diameter $d_{int}=101$ mm, inner height $h_{int}=95$ mm, wall thickness t=4.3 mm and 35 mm height $h_{a}$ from the bottom of the set up to the antenna.

At simulation stage, the ridged horn antennas in [6] integrated on the hexagonal set up in Figure 3, have been modeled using CST Studio. The phantom material filling the horns and the set up has been defined according to [16] ($\varepsilon_{r}^{\prime }=57$ and tanδ=0.6 at 0.8 GHz). It must be considered that the resonance frequency of the ID must fall within the operational bandwidth of the horns (0.5 GHz to 3 GHz [6]). Also, the ID must be aligned parallel to the vertical E-field radiated by the antennas for optimal detection.

In Figure 4 and Figure 5 the simulated joint E-field distribution of both horn antennas, transverse to the aperture plane of the antennas, is represented at 0.8 GHz (estimated resonance frequency of the considered ID) for the case when no implant is present in the scenario (Figure 4), and for an implanted device placed in front of the aperture of one of the antennas (Figure 5), clearly showing the influence of the implant. We can also observe the necessity of placing the probe antennas in such a way that the implant is located inside the crossing region of the field distribution radiated by both (Tx and Rx) antennas, which contributes to ID sensing.

The results exhibited in this section correspond to the simulation/experimental measurement of an alumina rod of 24 mm length and 4 mm diameter (parallel to the E-field radiated by the horn antennas) for different positions along the y axis (see Figure 3), as a mean to asses the variation introduced by the resonant object in the scattering matrix ([S]) of the Tx and Rx antennas. Baseline results (no Rod) for the medium without the ID are also included. These measurements exhibit a repeatability error, computed according to the standard deviation of the mean, below 0.04 within the frequency band of interest.

Figure 6 and Figure 7 represent the magnitude of the transmission coefficient ($S_{12}$) in (dB) simulated and measured respectively, as a function of frequency. Because of the high ohmic loss present in this kind of media (in the order of 2 dB/cm), it is possible to extract better information from the transmission (bistatic measurement) than from the reflection parameters (monostatic measurement). In Figure 8 and Figure 9 the results presented in the frequency domain have been time-converted applying the IFFT.

From the representation of the $S_{12}$ parameter for the baseline scenario (no ID), in both simulated and measured results, a steady slope with a maximum around 0.5 GHz is observed, which is in agreement with the results presented in [6]. When the ID is placed in front of the Tx and Rx, a modification in the $S_{12}$ in the shape of a “resonance” is produced. When the metallic rod is sequentially shifted, the frequency at which this “resonance” is produced and its depth variate too, as illustrated both in the simulation (Figure 6) and the ex-vivo measurement (Figure 7). The occurrence of this resonance indicates the capability to detect the presence of a resonant object implanted in a biological material. While the change of its frequency and depth triggered by a variation of its distance to the Tx and Rx antennas mean that it is also possible to locate the ID.

When these figures for the transmission coefficient are transformed to the time domain (Figure 8 and Figure 9), a more evident representation of the aforementioned phenomenon is observed. For the case where no ID is present in the scenario, it is possible to notice a single maximum in $IFFT(S_{12})$ which corresponds to the direct ray between the Tx and Rx EGRH antennas. However, when the ID is introduced, two maximums are observed. The first one corresponds to the direct ray (from Tx to Rx), while the second one corresponds to a reflected ray generated by the presence of the resonant metallic implant. It is of interest to notice that all direct rays peak at the same time instant, meanwhile, the reflected rays achieve their maximum later (at a time corresponding with its actual position) as they are placed further from the Tx/Rx antennas. The amplitude of the reflected ray decreases as the distance to the Tx/Rx antenna increases, as the amount of losses to cope with increase in the distance.

The 1.4 ns time shift between IFFT of the ex vivo experimental measurement and the CST simulation is associated to the time-delay introduced by the 0.3 m teflon coaxial cables employed at measurement stage.

## 4. In-Field In Vivo Measurement of Implanted Coronary Stent

After validating the applicability of the resonance-based microwave technique for body implant sensing with simulations and experimental measurements, we were able to test its performance in an in vivo scenario. A swine animal implanted with a coronary stent has been measured using the aforementioned procedure. Figure 10 depicts the display of the antennas on the side of the pig’s rib cage, aiming to the implanted stent, with a bistatic geometry equivalent to the one in Figure 3.

The propagation conditions present during the in vivo measurement, although similar, slightly variate with respect to the ex-vivo scenario. Before arriving to the implant, the signal transmitted by the Tx horn antenna must go through a layer of skin, fat and muscle, and then through the bone of the rib cage. This propagation conditions along with the lack of knowledge on the precise location and orientation of the stent, required a limited surface and angular scan. The flexibility and sensitivity of the technique allowed a proper disposition of the Tx/Rx probes which at the end permitted to obtain robust results.

From Figure 11, it is observed that we are able to detect the presence of the ID from the information in the magnitude of the transmission coefficient, while Figure 12 displays in the time domain the reflection produced by the presence of the stent. Based on the results of Figure 11 and Figure 12, and the geometry of the applied probes in Figure 10, an approximate depth of 3 cm with respect to the plane of the horn antennas may be estimated.

## 5. Conclusions

A novel technique for sensing of body implants has been analytically presented and validated through simulations, experimental measurements of a phantom mimicking the dielectric properties of biological tissue and, finally, by conducting an in vivo measurement on a coronary stent implanted in a swine animal model.

It is possible to extract information regarding the presence and location of medical implants from the resonant scattering produced by the device itself, with no need to make any modification in the implant.

From the results obtained during the experimental measurements, it is possible to say that the resonance-based microwave technique with the EGRH antennas has detected IDs up to 4 cm deep measured with a general purpose laboratory instrument. This range could be extended to 10 cm (we should not find IDs placed much deeper in real human body exploration) with an enhanced SNR (Signal to Noise Ratio) produced with the adequate featured instrumentation.

## Figures and Tables

**Figure 1 sensors-19-04828-f001:**
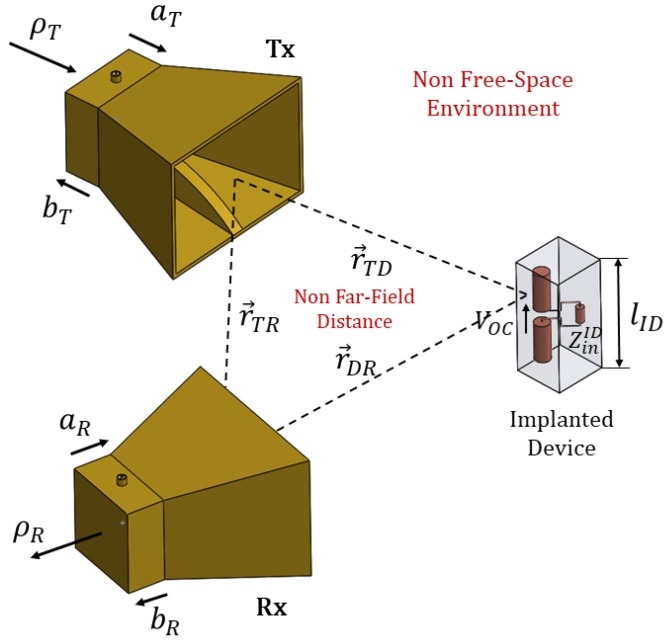
General Scenario.

**Figure 2 sensors-19-04828-f002:**
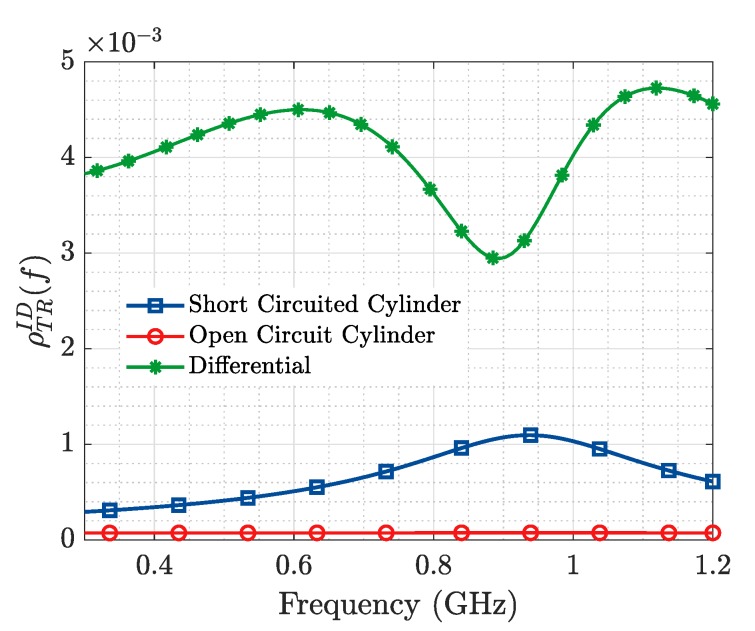
$\rho_{TR}^{ID}\left(f\right)$ of a 20 mm conducting cylinder representing the ID. The resonance is distinguished when the ID is fully operational (continuous object or $Z_{LD}^{ID}=0$), from a non-operational state (breakdown or $Z_{LD}^{ID}=1000$). The interferent differential signal is also reported.

**Figure 3 sensors-19-04828-f003:**
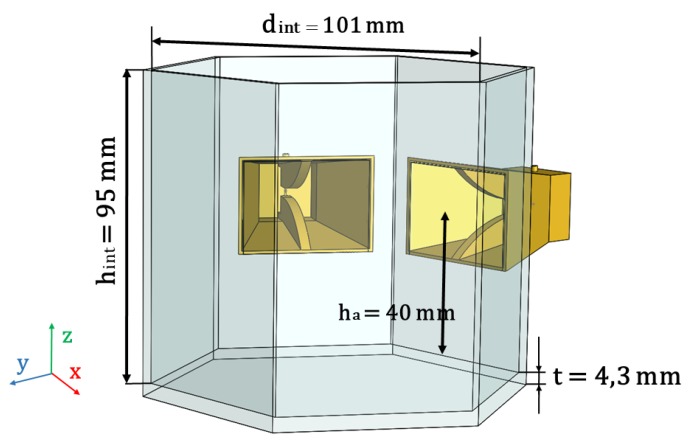
Schematic of the Experimental Measurement Set Up.

**Figure 4 sensors-19-04828-f004:**
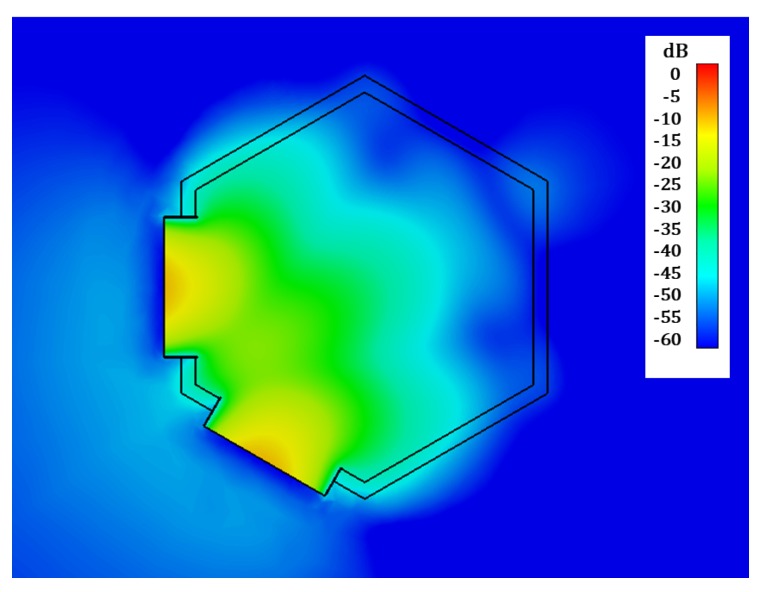
Simulated joint E-field distribution into the transverse plane of the horn antennas at 0.8 GHz. No ID.

**Figure 5 sensors-19-04828-f005:**
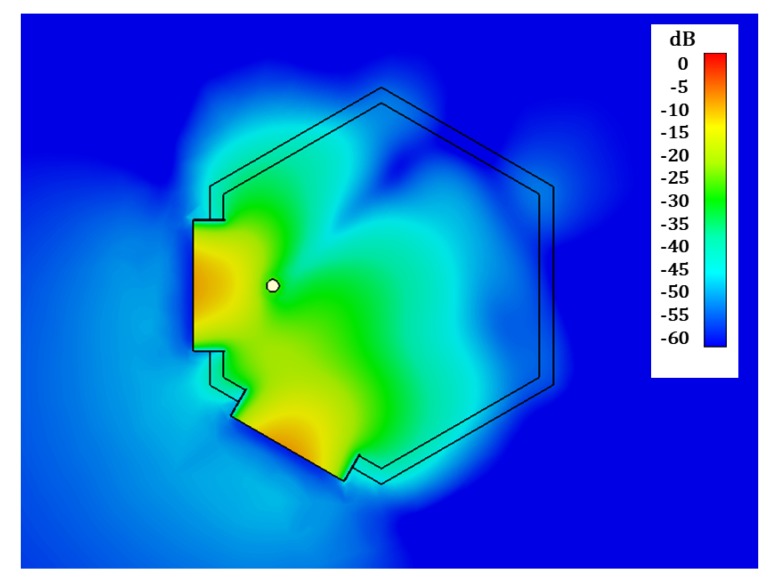
Simulated joint E-field distribution into the transverse plane of the horn antennas at 0.8 GHz. Frontal ID.

**Figure 6 sensors-19-04828-f006:**
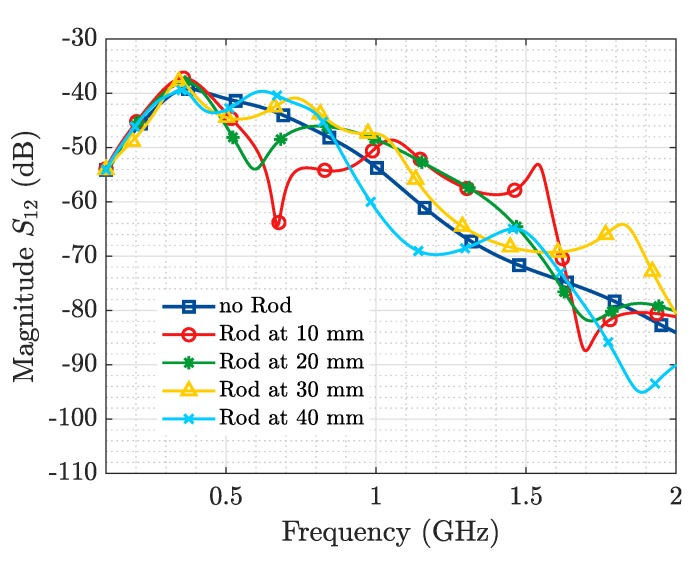
Metallic Rod Detection. Simulation $S_{12}$.

**Figure 7 sensors-19-04828-f007:**
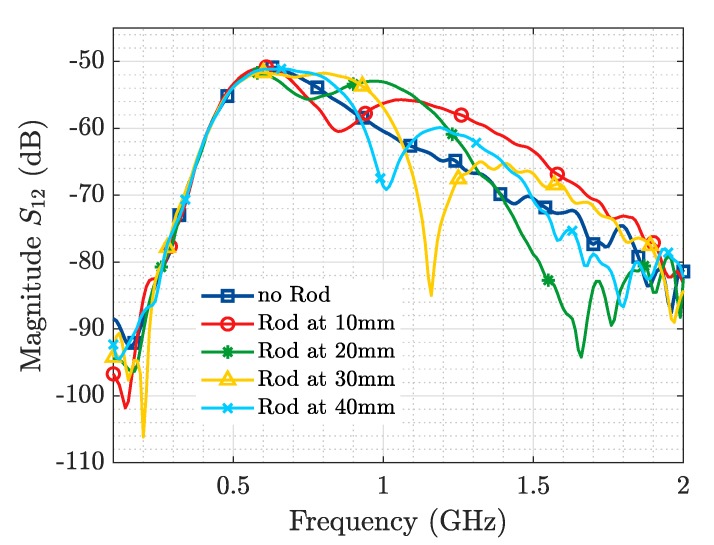
Metallic Rod Detection. Experimental measurement $S_{12}$.

**Figure 8 sensors-19-04828-f008:**
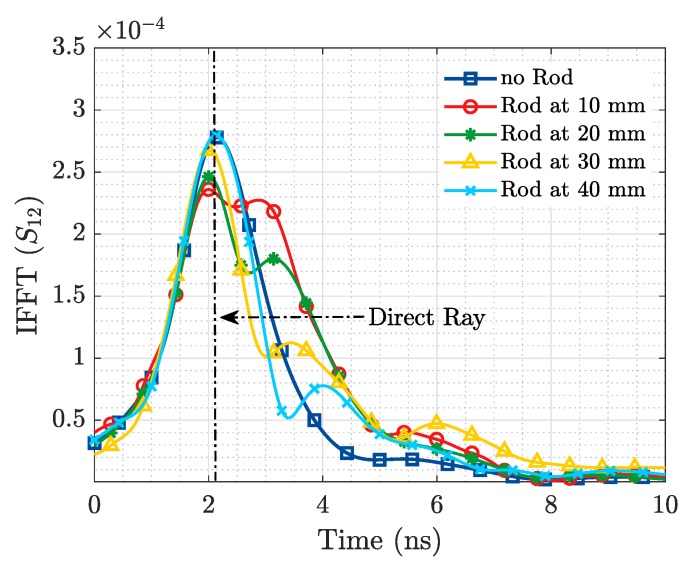
Metallic Rod Detection. Simulation IFFT.

**Figure 9 sensors-19-04828-f009:**
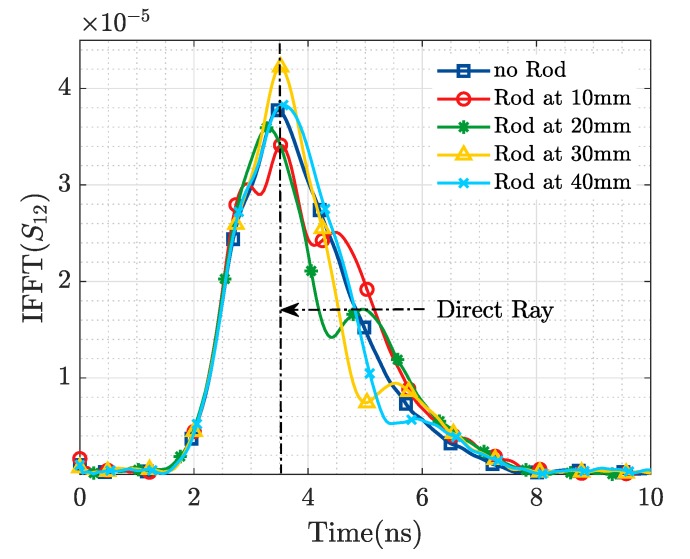
Metallic Rod Detection. Experimental measurement IFFT.

**Figure 10 sensors-19-04828-f010:**
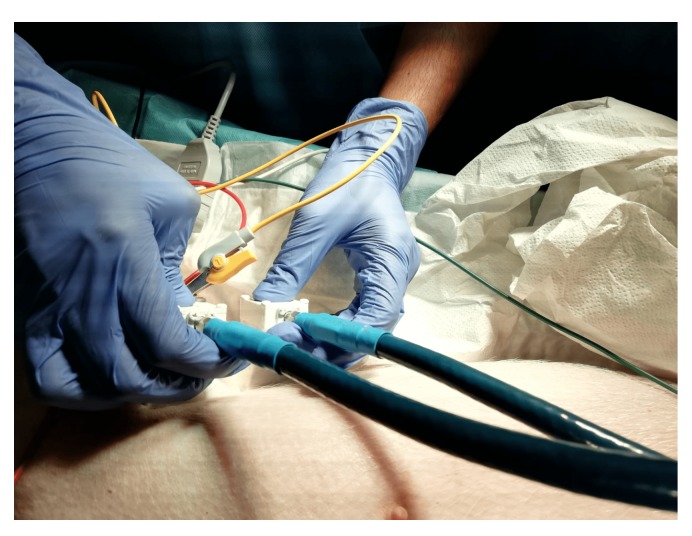
In vivo measurement.

**Figure 11 sensors-19-04828-f011:**
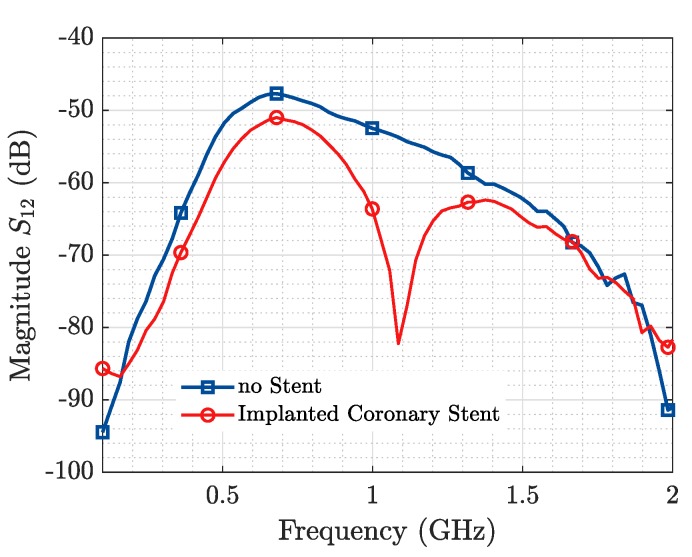
Coronary Stent Sensing. $S_{12}$.

**Figure 12 sensors-19-04828-f012:**
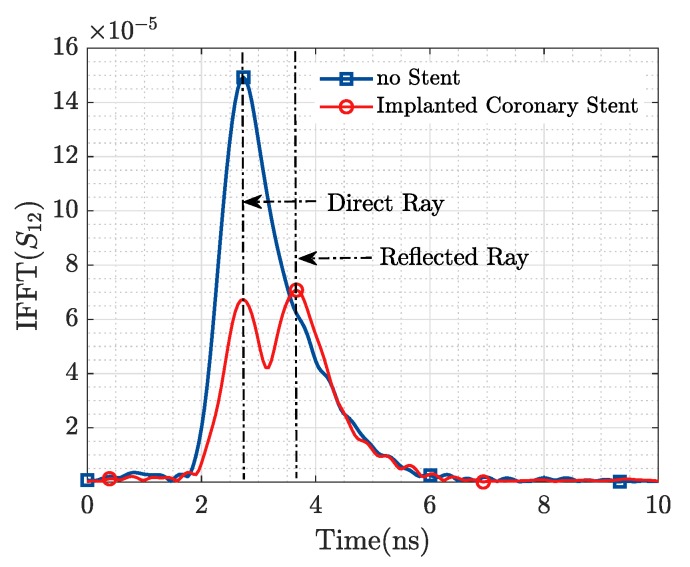
Coronary Stent Sensing. IFFT.

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
