# Peer review of "Resonance-Based Microwave Technique for Body Implant Sensing"

_sensors, 2019, doi:10.3390/s19224828_

Round 1
Reviewer 1 Report
Dear authors,
Thanks for submitting this manuscript, which presents a new sensing technique using resonance-based microwave.
I have carefully read your manuscript with great interest.
I think that it should sound very interesting for readers and this paper overall well written.
I have a few comments.
Part 3: Authors need to detail explain in simulation part including (simulation’s conditions & limitation) for the readers.
Part 4: In figures 8&9, How many times repeated the experimental measurement?
If authors have repeated data, it must be explaining with sensing’s repeat precision.
Also, it looked somewhat the difference of simulation & experimental data (time-delay). For explanation of the difference, authors need to more add the discussion in manuscript.
Generally, it seems too short the discussion for result in Part 3 & 4.
Sincerely,
Reviewer 2 Report
This is an interesting paper. You are proposing to use implants as signal scattering devices and detect changes in the devices as various factors influence the resonance frequency(s) of the device. It took me a while to realize this. Perhaps a clearer statement up front on the nature of what you are proposing would help.
The UHF based scattering approach could become a routine diagnostic check in the future for those with implants. It would be nice if data on implants in addition to the stints were presented, although your purpose here was to simply present the technique. I look forward to follow-up work that evaluates more potential applications.
